# Regularized Spectral Clustering under the Degree-Corrected Stochastic Blockmodel

**Tai Qin**
Department of Statistics
University of Wisconsin-Madison
Madison, WI
qin@stat.wisc.edu

**Karl Rohe**
Department of Statistics
University of Wisconsin-Madison
Madison, WI
karlrohe@stat.wisc.edu

## Abstract

Spectral clustering is a fast and popular algorithm for finding clusters in networks. Recently, Chaudhuri et al. [1] and Amini et al. [2] proposed inspired variations on the algorithm that artificially inflate the node degrees for improved statistical performance. The current paper extends the previous statistical estimation results to the more canonical spectral clustering algorithm in a way that removes any assumption on the minimum degree and provides guidance on the choice of the tuning parameter. Moreover, our results show how the "star shape" in the eigenvectors–a common feature of empirical networks–can be explained by the Degree-Corrected Stochastic Blockmodel and the Extended Planted Partition model, two statistical models that allow for highly heterogeneous degrees. Throughout, the paper characterizes and justifies several of the variations of the spectral clustering algorithm in terms of these models.

## 1 Introduction

Our lives are embedded in networks–social, biological, communication, etc.– and many researchers wish to analyze these networks to gain a deeper understanding of the underlying mechanisms. Some types of underlying mechanisms generate communities (aka clusters or modularities) in the network. As machine learners, our aim is not merely to devise algorithms for community detection, but also to study the algorithm's estimation properties, to understand if and when we can make justifiable inferences from the estimated communities to the underlying mechanisms. Spectral clustering is a fast and popular technique for finding communities in networks. Several previous authors have studied the estimation properties of spectral clustering under various statistical network models (McSherry [3], Dasgupta et al. [4], Coja-Oghlan and Lanka [5], Ames and Vavasis [6], Rohe et al. [7], Sussman et al. [8] and Chaudhuri et al. [1]). Recently, Chaudhuri et al. [1] and Amini et al. [2] proposed two inspired ways of artificially inflating the node degrees in ways that provide statistical regularization to spectral clustering.

This paper examines the statistical estimation performance of regularized spectral clustering under the Degree-Corrected Stochastic Blockmodel (DC-SBM), an extension of the Stochastic Blockmodel (SBM) that allows for heterogeneous degrees (Holland and Leinhardt [9], Karrer and Newman [10]). The SBM and the DC-SBM are closely related to the planted partition model and the extended planted partition model, respectively. We extend the previous results in the following ways: (a) In contrast to previous studies, this paper studies the regularization step with a canonical version of spectral clustering that uses k-means. The results do not require any assumptions on the minimum expected node degree; instead, there is a threshold demonstrating that higher degree nodes are easier to cluster. This threshold is a function of the leverage scores that have proven essential in other contexts, for both graph algorithms and network data analysis (see Mahoney [11] and references therein). These are the first results that relate leverage scores to the statistical performance

of spectral clustering. (b) This paper provides more guidance for data analytic issues than previous approaches. First, the results suggest an appropriate range for the regularization parameter. Second, our analysis gives a (statistical) model-based explanation for the "star-shaped" figure that often appears in empirical eigenvectors. This demonstrates how projecting the rows of the eigenvector matrix onto the unit sphere (an algorithmic step proposed by Ng et al. [12]) removes the ancillary effects of heterogeneous degrees under the DC-SBM. Our results highlight when this step may be unwise.

**Preliminaries:** Throughout, we study undirected and unweighted graphs or networks. Define a graph as $G(E, V)$, where $V = \{v_1, v_2, \ldots, v_N\}$ is the vertex or node set and $E$ is the edge set. We will refer to node $v_i$ as node $i$. $E$ contains a pair $(i, j)$ if there is an edge between node $i$ and $j$. The edge set can be represented by the adjacency matrix $A \in \{0, 1\}^{n \times n}$. $A_{ij} = A_{ji} = 1$ if $(i, j)$ is in the edge set and $A_{ij} = A_{ji} = 0$ otherwise. Define the diagonal matrix $D$ and the normalized Graph Laplacian $L$, both elements of $\mathcal{R}^{N \times N}$, in the following way:

$$D_{ii} = \sum_j A_{ij}, \qquad L = D^{-1/2} A D^{-1/2}.$$

The following notations will be used throughout the paper: $||\cdot||$ denotes the spectral norm, and $||\cdot||_F$ denotes the Frobenius norm. For two sequence of variables $\{x_N\}$ and $\{y_N\}$, we say $x_N = \omega(y_N)$ if and only if $y_N/x_N = o(1)$. $\delta_{(.,.)}$ is the indicator function where $\delta_{x,y} = 1$ if $x = y$ and $\delta_{x,y} = 0$ if $x \neq y$.

## 2 The Algorithm: Regularized Spectral Clustering (RSC)

For a sparse network with strong degree heterogeneity, standard spectral clustering often fails to function properly (Amini et al. [2], Jin [13]). To account for this, Chaudhuri et al. [1] proposed the regularized graph Laplacian that can be defined as

$$L_\tau = D_\tau^{-1/2} A D_\tau^{-1/2} \in \mathcal{R}^{N \times N}$$

where $D_\tau = D + \tau I$ for $\tau \geq 0$.

The spectral algorithm proposed and studied by Chaudhuri et al. [1] divides the nodes into two random subsets and only uses the induced subgraph on one of those random subsets to compute the spectral decomposition. In this paper, we will study the more traditional version of spectral algorithm that uses the spectral decomposition on the entire matrix (Ng et al. [12]). Define the regularized spectral clustering (RSC) algorithm as follows:

1. Given input adjacency matrix $A$, number of clusters $K$, and regularizer $\tau$, calculate the regularized graph Laplacian $L_\tau$. (As discussed later, a good default for $\tau$ is the average node degree.)

2. Find the eigenvectors $X_1, ..., X_K \in \mathcal{R}^N$ corresponding to the $K$ largest eigenvalues of $L_\tau$. Form $X = [X_1, ..., X_K] \in \mathcal{R}^{N \times K}$ by putting the eigenvectors into the columns.

3. Form the matrix $X^* \in \mathcal{R}^{N \times K}$ from $X$ by normalizing each of $X$'s rows to have unit length. That is, project each row of X onto the unit sphere of $\mathcal{R}^K$ ($X_{ij}^* = X_{ij}/(\sum_j X_{ij}^2)^{1/2}$).

4. Treat each row of $X^*$ as a point in $\mathcal{R}^K$, and run k-means with $K$ clusters. This creates $K$ non-overlapping sets $V_1, ..., V_K$ whose union is V.

5. Output $V_1, ..., V_K$. Node $i$ is assigned to cluster $r$ if the $i$'th row of $X^*$ is assigned to $V_r$.

This paper will refer to "standard spectral clustering" as the above algorithm with $L$ replacing $L_\tau$.

These spectral algorithms have two main steps: 1) find the principal eigenspace of the (regularized) graph Laplacian; 2) determine the clusters in the low dimensional eigenspace. Later, we will study RSC under the Degree-Corrected Stochastic Blockmodel and show rigorously how regularization helps to maintain cluster information in step (a) and why normalizing the rows of $X$ helps in step (b). From now on, we use $X_\tau$ and $X_\tau^*$ instead of $X$ and $X^*$ to emphasize that they are related to $L_\tau$. Let $X_\tau^i$ and $[X_\tau^*]^i$ denote the $i$'th row of $X_\tau$ and $X_\tau^*$.

The next section introduces the Degree-Corrected Stochastic Blockmodel and its matrix formulation.

# 3 The Degree-Corrected Stochastic Blockmodel (DC-SBM)

In the Stochastic Blockmodel (SBM), each node belongs to one of $K$ blocks. Each edge corresponds to an independent Bernoulli random variable where the probability of an edge between any two nodes depends only on the block memberships of the two nodes (Holland and Leinhardt [9]). The formal definition is as follows.

**Definition 3.1.** *For a node set $\{1, 2, ..., N\}$, let $z : \{1, 2, ..., N\} \rightarrow \{1, 2, ..., K\}$ partition the $N$ nodes into $K$ blocks. So, $z_i$ equals the block membership for node $i$. Let $\mathbf{B}$ be a $K \times K$ matrix where $\mathbf{B}_{ab} \in [0, 1]$ for all $a, b$. Then under the SBM, the probability of an edge between $i$ and $j$ is $P_{ij} = P_{ji} = \mathbf{B}_{z_i z_j}$ for any $i, j = 1, 2, ..., n$. Given $z$, all edges are independent.*

One limitation of the SBM is that it presumes all nodes within the same block have the same expected degree. The Degree-Corrected Stochastic Blockmodel (DC-SBM) (Karrer and Newman [10]) is a generalization of the SBM that adds an additional set of parameters ($\theta_i > 0$ for each node $i$) that control the node degrees. Let $\mathbf{B}$ be a $K \times K$ matrix where $\mathbf{B}_{ab} \geq 0$ for all $a, b$. Then the probability of an edge between node $i$ and node $j$ is $\theta_i \theta_j \mathbf{B}_{z_i z_j}$, where $\theta_i \theta_j \mathbf{B}_{z_i z_j} \in [0, 1]$ for any $i, j = 1, 2, ..., n$. Parameters $\theta_i$ are arbitrary to within a multiplicative constant that is absorbed into $\mathbf{B}$. To make it identifiable, Karrer and Newman [10] suggest imposing the constraint that, within each block, the summation of $\theta_i$'s is 1. That is, $\sum_i \theta_i \delta_{z_i, r} = 1$ for any block label $r$. Under this constraint, $\mathbf{B}$ has explicit meaning: If $s \neq t$, $\mathbf{B_{st}}$ represents the expected number of links between block $s$ and block $t$ and if $s = t$, $\mathbf{B_{st}}$ is twice the expected number of links within block $s$. Throughout the paper, we assume that $\mathbf{B}$ is positive definite.

Under the DC-SBM, define $\mathscr{A} \triangleq \mathbb{E}A$. This matrix can be expressed as a product of the matrices,

$$\mathscr{A} = \Theta Z \mathbf{B} Z^T \Theta,$$

where (1) $\Theta \in \mathcal{R}^{N \times N}$ is a diagonal matrix whose $ii$'th element is $\theta_i$ and (2) $Z \in \{0, 1\}^{N \times K}$ is the membership matrix with $Z_{it} = 1$ if and only if node $i$ belongs to block $t$ (i.e. $z_i = t$).

## 3.1 Population Analysis

Under the DC-SBM, if the partition is identifiable, then one should be able to determine the partition from $\mathscr{A}$. This section shows that with the population adjacency matrix $\mathscr{A}$ and a proper regularizer $\tau$, RSC perfectly reconstructs the block partition.

Define the diagonal matrix $\mathscr{D}$ to contain the expected node degrees, $\mathscr{D}_{ii} = \sum_j \mathscr{A}_{ij}$ and define $\mathscr{D}_\tau = \mathscr{D} + \tau I$ where $\tau \geq 0$ is the regularizer. Then, define the population graph Laplacian $\mathscr{L}$ and the population version of regularized graph Laplacian $\mathscr{L}_\tau$, both elements of $\mathcal{R}^{N \times N}$, in the following way:

$$\mathscr{L} = \mathscr{D}^{-1/2} \mathscr{A} \mathscr{D}^{-1/2}, \qquad \mathscr{L}_\tau = \mathscr{D}_\tau^{-1/2} \mathscr{A} \mathscr{D}_\tau^{-1/2}.$$

Define $D_B \in \mathcal{R}^{K \times K}$ as a diagonal matrix whose $(s, s)$'th element is $[D_B]_{ss} = \sum_t B_{st}$. A couple lines of algebra shows that $[D_B]_{ss} = W_s$ is the total expected degrees of nodes from block $s$ and that $\mathscr{D}_{ii} = \theta_i [D_B]_{z_i z_i}$. Using these quantities, the next Lemma gives an explicit form for $\mathscr{L}_\tau$ as a product of the parameter matrices.

**Lemma 3.2.** *(Explicit form for $\mathscr{L}_\tau$) Under the DC-SBM with $K$ blocks with parameters $\{\mathbf{B}, Z, \Theta\}$, define $\theta_i^\tau$ as:*

$$\theta_i^\tau = \frac{\theta_i^2}{\theta_i + \tau/W_{z_i}} = \theta_i \frac{\mathscr{D}_{ii}}{\mathscr{D}_{ii} + \tau}.$$

*Let $\Theta_\tau \in \mathcal{R}^{n \times n}$ be a diagonal matrix whose $ii$'th entry is $\theta_i^\tau$. Define $B_L = D_B^{-1/2} B D_B^{-1/2}$, then $\mathscr{L}_\tau$ can be written*

$$\mathscr{L}_\tau = \mathscr{D}_\tau^{-\frac{1}{2}} \mathscr{A} \mathscr{D}_\tau^{-\frac{1}{2}} = \Theta_\tau^{\frac{1}{2}} Z B_L Z^T \Theta_\tau^{\frac{1}{2}}.$$

Recall that $\mathscr{A} = \Theta Z \mathbf{B} Z^T \Theta$. Lemma 3.2 demonstrates that $\mathscr{L}_\tau$ has a similarly simple form that separates the block-related information ($B_L$) and node specific information ($\Theta_\tau$). Notice that if $\tau = 0$, then $\Theta_0 = \Theta$ and $\mathscr{L} = \mathscr{D}^{-\frac{1}{2}} \mathscr{A} \mathscr{D}^{-\frac{1}{2}} = \Theta^{\frac{1}{2}} Z B_L Z^T \Theta^{\frac{1}{2}}$. The next lemma shows that $\mathscr{L}_\tau$ has rank $K$ and describes how its eigen-decomposition can be expressed in terms of $Z$ and $\Theta$.

**Lemma 3.3.** *(Eigen-decomposition for $\mathscr{L}_\tau$) Under the DC-SBM with $K$ blocks and parameters $\{\mathbf{B}, Z, \Theta\}$, $\mathscr{L}_\lambda$ has $K$ positive eigenvalues. The remaining $N - K$ eigenvalues are zero. Denote the $K$ positive eigenvalues of $\mathscr{L}_\tau$ as $\lambda_1 \geq \lambda_2 \geq ... \geq \lambda_K > 0$ and let $\mathscr{X}_\tau \in \mathcal{R}^{N \times K}$ contain the eigenvector corresponding to $\lambda_i$ in its $i$'th column. Define $\mathscr{X}_\tau^*$ to be the row-normalized version of $\mathscr{X}_\tau$, similar to $X_\tau^*$ as defined in the RSC algorithm in Section 2. Then, there exists an orthogonal matrix $U \in \mathcal{R}^{K \times K}$ depending on $\tau$, such that*

1. *$\mathscr{X}_\tau = \Theta_\tau^{\frac{1}{2}} Z (Z^T \Theta_\tau Z)^{-1/2} U$*

2. *$\mathscr{X}_\tau^* = ZU$, $Z_i \neq Z_j \Leftrightarrow Z_i U \neq Z_j U$, where $Z_i$ denote the $i$'th row of the membership matrix $Z$.*

This lemma provides four useful facts about the matrices $\mathscr{X}_\tau$ and $\mathscr{X}_\tau^*$. First, if two nodes $i$ and $j$ belong to the same block, then the corresponding rows of $\mathscr{X}_\tau$ (denoted as $\mathscr{X}_\tau^i$ and $\mathscr{X}_\tau^j$) both point in the same direction, but with different lengths: $||\mathscr{X}_\tau^i||_2 = (\frac{\theta_i^\tau}{\sum_j \theta_j^\tau \delta_{z_j, z_i}})^{1/2}$. Second, if two nodes $i$ and $j$ belong to different blocks, then $\mathscr{X}_\tau^i$ and $\mathscr{X}_\tau^j$ are orthogonal to each other. Third, if $z_i = z_j$ then after projecting these points onto the sphere as in $\mathscr{X}_\tau^*$, the rows are equal: $[\mathscr{X}_\tau^*]^i = [\mathscr{X}_\tau^*]^j = U_{z_i}$. Finally, if $z_i \neq z_j$, then the rows are perpendicular, $[\mathscr{X}_\tau^*]^i \perp [\mathscr{X}_\tau^*]^j$. Figure 1 illustrates the geometry of $\mathscr{X}_\tau$ and $\mathscr{X}_\tau^*$ when there are three underlying blocks. Notice that running k-means on the rows of $\mathscr{X}_\lambda^*$ (in right panel of Figure 1) will return perfect clusters.

Note that if $\Theta$ were the identity matrix, then the left panel in Figure 1 would look like the right panel in Figure 1; without degree heterogeneity, there would be no star shape and no need for a projection step. This suggests that the star shaped figure often observed in data analysis stems from the degree heterogeneity in the network.

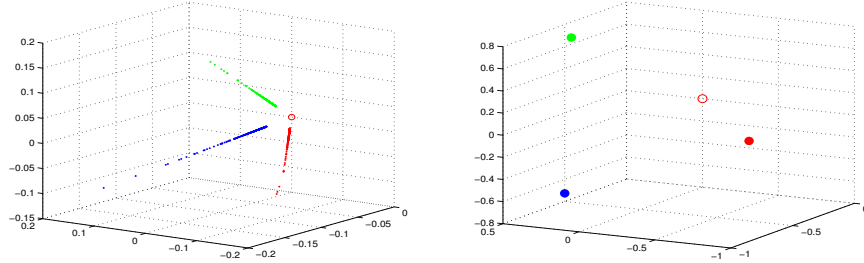

Figure 1: In this numerical example, $\mathscr{A}$ comes from the DC-SBM with three blocks. Each point corresponds to one row of the matrix $\mathscr{X}_\tau$ (in left panel) or $\mathscr{X}_\tau^*$ (in right panel). The different colors correspond to three different blocks. The hollow circle is the origin. Without normalization (left panel), the nodes with same block membership share the **same direction** in the projected space. After normalization (right panel), nodes with same block membership share the **same position** in the projected space.

## 4 Regularized Spectral Clustering with the Degree Corrected model

This section bounds the mis-clustering rate of Regularized Spectral Clustering under the DC-SBM. The section proceeds as follows: Theorem 4.1 shows that $L_\tau$ is close to $\mathscr{L}_\tau$. Theorem 4.2 shows that $X_\tau$ is close to $\mathscr{X}_\tau$ and that $X_\tau^*$ is close to $\mathscr{X}_\tau^*$. Finally, Theorem 4.4 shows that the output from RSC with $L_\tau$ is close to the true partition in the DC-SBM (using Lemma 3.3).

**Theorem 4.1.** *(Concentration of the regularized Graph Laplacian) Let $G$ be a random graph, with independent edges and $pr(v_i \sim v_j) = p_{ij}$. Let $\delta$ be the minimum expected degree of $G$, that is $\delta = \min_i \mathscr{D}_{ii}$. For any $\epsilon > 0$, if $\delta + \tau > 3 \ln N + 3 \ln(4/\epsilon)$, then with probability at least $1 - \epsilon$,*

$$||L_\tau - \mathscr{L}_\tau|| \leq 4\sqrt{\frac{3 \ln(4N/\epsilon)}{\delta + \tau}}. \tag{1}$$

**Remark:** This theorem builds on the results of Chung and Radcliffe [14] and Chaudhuri et al. [1] which give a seemingly similar bound on $||L - \mathscr{L}||$ and $||D_\tau^{-1}A - \mathscr{D}_\tau^{-1}\mathscr{A}||$. However, the previous papers require that $\delta \geq c \ln N$, where $c$ is some constant. This assumption is not satisfied in a large proportion of sparse empirical networks with heterogeneous degrees. In fact, the regularized graph Laplacian is most interesting when this condition fails, i.e. when there are several nodes with very low degrees. Theorem 4.1 only assumes that $\delta + \tau > 3 \ln N + 3 \ln(4/\epsilon)$. This is the fundamental reason that RSC works for networks containing some nodes with extremely small degrees. It shows that, by introducing a proper regularizer $\tau$, $||L_\tau - \mathscr{L}_\tau||$ can be well bounded, even with $\delta$ very small. Later we will show that a suitable choice of $\tau$ is the average degree.

The next theorem bounds the difference between the empirical and population eigenvectors (and their row normalized versions) in terms of the Frobenius norm.

**Theorem 4.2.** *Let A be the adjacency matrix generated from the DC-SBM with $K$ blocks and parameters $\{\mathbf{B}, Z, \Theta\}$. Let $\lambda_1 \geq \lambda_2 \geq ... \geq \lambda_K > 0$ be the only $K$ positive eigenvalues of $\mathscr{L}_\tau$. Let $X_\tau$ and $\mathscr{X}_\tau \in \mathcal{R}^{N \times K}$ contain the top $K$ eigenvectors of $L_\tau$ and $\mathscr{L}_\tau$ respectively. Define $m = \min_i\{\min\{||X_\tau^i||_2, ||\mathscr{X}_\tau^i||_2\}\}$ as the length of the shortest row in $X_\tau$ and $\mathscr{X}_\tau$. Let $X_\tau^*$ and $\mathscr{X}_\tau^* \in \mathcal{R}^{N \times K}$ be the row normalized versions of $X_\tau$ and $\mathscr{X}_\tau$, as defined in step 3 of the RSC algorithm.*

*For any $\epsilon > 0$ and sufficiently large $N$, assume that*

$$(a) \ \sqrt{\frac{K \ln(4N/\epsilon)}{\delta + \tau}} \leq \frac{1}{8\sqrt{3}}\lambda_K, \qquad (b) \ \delta + \tau > 3 \ln N + 3 \ln(4/\epsilon),$$

*then with probability at least $1 - \epsilon$, the following holds,*

$$||X_\tau - \mathscr{X}_\tau\mathcal{O}||_F \leq c_0 \frac{1}{\lambda_K}\sqrt{\frac{K \ln(4N/\epsilon)}{\delta + \tau}}, \ \ and \ \ ||X_\tau^* - \mathscr{X}_\tau^*\mathcal{O}||_F \leq c_0 \frac{1}{m\lambda_K}\sqrt{\frac{K \ln(4N/\epsilon)}{\delta + \tau}}. \ (2)$$

The proof of Theorem 4.2 can be found in the supplementary materials.

Next we use Theorem 4.2 to derive a bound on the mis-clustering rate of RSC. To define "mis-clustered", recall that RSC applies the k-means algorithm to the rows of $X_\tau^*$, where each row is a point in $\mathcal{R}^K$. Each row is assigned to one cluster, and each of these clusters has a centroid from k-means. Define $C_1, \ldots, C_n \in \mathcal{R}^K$ such that $C_i$ is the centroid corresponding to the $i$'th row of $X_\tau^*$. Similarly, run k-means on the rows of the population eigenvector matrix $\mathscr{X}_\tau^*$ and define the population centroids $\mathcal{C}_1, \ldots, \mathcal{C}_n \in \mathcal{R}^K$. In essence, we consider node $i$ correctly clustered if $C_i$ is closer to $\mathcal{C}_i$ than it is to any other $\mathcal{C}_j$ for all $j$ with $Z_j \neq Z_i$.

The definition is complicated by the fact that, if any of the $\lambda_1, \ldots, \lambda_K$ are equal, then only the subspace spanned by their eigenvectors is identifiable. Similarly, if any of those eigenvalues are close together, then the estimation results for the individual eigenvectors are much worse that for the estimation results for the subspace that they span. Because clustering only requires estimation of the correct subspace, our definition of correctly clustered is amended with the rotation $\mathcal{O}^T \in R^{K \times K}$, the matrix which minimizes $||X_\tau^*\mathcal{O}^T - \mathscr{X}_\tau^*||_F$. This is referred to as the orthogonal Procrustes problem and [15] shows how the singular value decomposition gives the solution.

**Definition 4.3.** *If $C_i\mathcal{O}^T$ is closer to $\mathcal{C}_i$ than it is to any other $\mathcal{C}_j$ for $j$ with $Z_j \neq Z_i$, then we say that node $i$ is correctly clustered. Define the set of mis-clustered nodes:*

$$\mathscr{M} = \{i : \exists j \neq i, s.t. ||C_i\mathcal{O}^T - \mathcal{C}_i||_2 > ||C_i\mathcal{O}^T - \mathcal{C}_j||_2\}. \qquad (3)$$

The next theorem bounds the mis-clustering rate $|\mathscr{M}|/N$.

**Theorem 4.4.** *(**Main Theorem**) Suppose $A \in \mathcal{R}^{N \times N}$ is an adjacency matrix of a graph $G$ generated from the DC-SBM with $K$ blocks and parameters $\{\mathbf{B}, Z, \Theta\}$. Let $\lambda_1 \geq \lambda_2 \geq ... \geq \lambda_K > 0$ be the $K$ positive eigenvalues of $\mathscr{L}_\tau$. Define $\mathscr{M}$, the set of mis-clustered nodes, as in Definition 4.3. Let $\delta$ be the minimum expected degree of $G$. For any $\epsilon > 0$ and sufficiently large $N$, assume (a) and (b) as in Theorem 4.2. Then with probability at least $1 - \epsilon$, the mis-clustering rate of RSC with regularization constant $\tau$ is bounded,*

$$|\mathscr{M}|/N \leq c_1\frac{K \ln(N/\epsilon)}{Nm^2(\delta + \tau)\lambda_K^2}. \qquad (4)$$

**Remark 1 (Choice of $\tau$):** The quality of the bound in Theorem 4.4 depends on $\tau$ through three terms: $(\delta + \tau)$, $\lambda_K$, and $m$. Setting $\tau$ equal to the average node degree balances these terms. In essence, if $\tau$ is too small, there is insufficient regularization. Specifically, if the minimum expected degree $\delta = O(\ln N)$, then we need $\tau \geq c(\epsilon) \ln N$ to have enough regularization to satisfy condition (b) on $\delta + \tau$. Alternatively, if $\tau$ is too large, it washes out significant eigenvalues.

To see that $\tau$ should not be too large, note that

$$C = (Z^T \Theta_\tau Z)^{1/2} B_L (Z^T \Theta_\tau Z)^{1/2} \in \mathcal{R}^{K \times K} \tag{5}$$

has the same eigenvalues as the largest $K$ eigenvalues of $\mathscr{L}_\tau$ (see supplementary materials for details). The matrix $Z^T \Theta_\tau Z$ is diagonal and the $(s, s)$'th element is the summation of $\theta_i^\tau$ within block $s$. If $\mathbb{E}M = \omega(N \ln N)$ where $M = \sum_i D_{ii}$ is the sum of the node degrees, then $\tau = \omega(M/N)$ sends the smallest diagonal entry of $Z^T \Theta_\tau Z$ to 0, sending $\lambda_K$, the smallest eigenvalue of $C$, to zero.

The trade-off between these two suggests that a proper range of $\tau$ is $(\alpha \frac{\mathbb{E}M}{N}, \beta \frac{\mathbb{E}M}{N})$, where $0 < \alpha < \beta$ are two constants. Keeping $\tau$ within this range guarantees that $\lambda_K$ is lower bounded by some constant depending only on $K$. In simulations, we find that $\tau = M/N$ (i.e. the average node degree) provides good results. The theoretical results only suggest that this is the correct rate. So, one could adjust this by a multiplicative constant. Our simulations suggest that the results are not sensitive to such adjustments.

**Remark 2 (Thresholding $m$):** Mahoney [11] (and references therein) shows how the leverage scores of $A$ and $L$ are informative for both data analysis and algorithmic stability. For $L$, the leverage score of node $i$ is $||X^i||_2^2$, the length of the $i$th row of the matrix containing the top $K$ eigenvectors. Theorem 4.4 is the first result that explicitly relates the leverage scores to the statistical performance of spectral clustering. Recall that $m^2$ is the minimum of the squared row lengths in $\mathscr{X}_\tau$ and $X_\tau$, that is the minimum leverage score in both $\mathscr{L}_\tau$ and $L_\tau$. This appears in the denominator of (4). The leverage scores in $\mathscr{L}_\tau$ have an explicit form $||\mathscr{X}_\tau^i||_2^2 = \frac{\theta_i^\tau}{\sum_j \theta_j^\tau \delta_{z_j, z_i}}$. So, if node $i$ has small expected degree, then $\theta_i^\tau$ is small, rendering $||\mathscr{X}_\tau^i||_2$ small. This can deteriorate the bound in Theorem 4.4. The problem arises from projecting $X_\tau^i$ onto the unit sphere for a node $i$ with small leverage; it amplifies a noisy measurement. Motivated by this intuition, the next corollary focuses on the high leverage nodes. More specifically, let $m^*$ denote the threshold. Define $S$ to be a subset of nodes whose leverage scores in $\mathscr{L}_\tau$ and $X_\tau$, $||\mathscr{X}_\tau^i||$ and $||X_\tau^i||$ exceed the threshold $m^*$:

$$S = \{i : ||\mathscr{X}_\tau^i|| \geq m^*, ||X_\tau^i|| \geq m^*\}.$$

Then by applying k-means on the set of vectors $\{[X_\tau^*]^i, i \in S\}$, we cluster these nodes. The following corollary bounds the mis-clustering rate on $S$.

**Corollary 4.5.** *Let $N_1 = |S|$ denote the number of nodes in $S$ and define $\mathscr{M}_1 = \mathscr{M} \cap S$ as the set of mis-clustered nodes restricted in $S$. With the same settings and assumptions as in Theorem 4.4, let $\gamma > 0$ be a constant and set $m^* = \gamma/\sqrt{N}$. If $N/N_1 = O(1)$, then by applying k-means on the set of vectors $\{[X_\tau^*]^i, i \in S\}$, we have with probability at least $1 - \epsilon$, there exist constant $c_2$ independent of $\epsilon$, such that*

$$|\mathscr{M}_1|/N_1 \leq c_2 \frac{K \ln(N_1/\epsilon)}{\gamma^2 (\delta + \tau) \lambda_K^2}. \tag{6}$$

In the main theorem (Theorem 4.4), the denominator of the upper bound contains $m^2$. Since we do not make a minimum degree assumption, this value potentially approaches zero, making the bound useless. Corollary 4.5 replaces $Nm^2$ with the constant $\gamma^2$, providing a superior bound when there are several small leverage scores.

If $\lambda_K$ (the $K$th largest eigenvalue of $\mathscr{L}_\tau$) is bounded below by some constant and $\tau = \omega(\ln N)$, then Corollary 4.5 implies that $|\mathscr{M}_1|/N_1 = o_p(1)$. The above thresholding procedure only clusters the nodes in $S$. To cluster all of the nodes, define the thresholded RSC (t-RSC) as follows:

(a) Follow step (1), (2), and (3) of RSC as in section 2.

(b) Apply k-means with $K$ clusters on the set S = $\{i, ||X_\tau^i||_2 \geq \gamma/\sqrt{N}\}$ and assign each of them to one of $V_1, ..., V_K$. Let $C_1, ..., C_K$ denote the $K$ centroids given by k-means.

(c) For each node $i \notin S$, find the centroid $C_s$ such that $||[X_\tau^*]^i - C_s||_2 = min_{1 \leq t \leq K} ||[X_\tau^*]^i - C_t||_2$. Assign node $i$ to $V_s$. Output $V_1, ... V_K$.

**Remark 3 (Applying to SC):** Theorem 4.4 can be easily applied to the standard SC algorithm under both the SBM and the DC-SBM by setting $\tau = 0$. In this setting, Theorem 4.4 improves upon the previous results for spectral clustering.

Define the four parameter Stochastic Blockmodel $SBM(p, r, s, K)$ as follows: $p$ is the probability of an edge occurring between two nodes from the same block, $r$ is the probability of an out-block linkage, $s$ is the number of nodes within each block, and $K$ is the number of blocks.

Because the SBM lacks degree heterogeneity within blocks, the rows of $\mathscr{X}$ within the same block already share the same length. So, it is not necessary to project $X^i$'s to the unit sphere. Under the four parameter model, $\lambda_K = (K[r/(p-r)] + 1)^{-1}$ (Rohe et al. [7]). Using Theorem 4.4, with $p$ and $r$ fixed and $p > r$, and applying k-means to the rows of $X$, we have

$$|\mathscr{M}|/N = O_p\left(\frac{K^2 \ln N}{N}\right). \tag{7}$$

If $K = o(\sqrt{\frac{N}{\ln N}})$, then $|\mathscr{M}|/N \to 0$ in probability. This improves the previous results that required $K = o(N^{1/3})$ (Rohe et al. [7]). Moreover, it makes the results for spectral clustering comparable to the results for the MLE in Choi et al. [16].

## 5 Simulation and Analysis of Political Blogs

This section compares five different methods of spectral clustering. Experiment 1 generates networks from the DC-SBM with a power-law degree distribution. Experiment 2 generates networks from the standard SBM. Finally, the benefits of regularization are illustrated on an empirical network from the political blogosphere during the 2004 presidential election (Adamic and Glance [17]).

The simulations compare (1) standard spectral clustering (SC), (2) RSC as defined in section 2, (3) RSC without projecting $X_\tau$ onto unit sphere (RSC_wp), (4) regularized SC with thresholding (t-RSC), and (5) spectral clustering with perturbation (SCP) (Amini et al. [2]) which applies SC to the perturbed adjacency matrix $A_{per} = A + a11^T$. In addition, experiment 2 compares the performance of RSC on the subset of nodes with high leverage scores (RSC on $S$) with the other 5 methods. We set $\tau = M/N$, threshold parameter $\gamma = 1$, and $a = M/N^2$ except otherwise specified.

**Experiment 1.** This experiment examines how degree heterogeneity affects the performance of the spectral clustering algorithms. The $\Theta$ parameters (from the DC-SBM) are drawn from the power law distribution with lower bound $x_{min} = 1$ and shape parameter $\beta \in \{2, 2.25, 2.5, 2.75, 3, 3.25, 3.5\}$. A smaller $\beta$ indicates to greater degree heterogeneity. For each fixed $\beta$, thirty networks are sampled. In each sample, $K = 3$ and each block contains 300 nodes ($N = 900$). Define the signal to noise ratio to be the expected number of in-block edges divided by the expected number of out-block edges. Throughout the simulations, the SNR is set to three and the expected average degree is set to eight.

The left panel of Figure 2 plots $\beta$ against the misclustering rate for SC, RSC, RSC_wp, t-RSC, SCP and RSC on $S$. Each point is the average of 30 sampled networks. Each line represents one method. If a method assigns more than $95\%$ of the nodes into one block, then we consider all nodes to be misclustered. The experiment shows that (1) if the degrees are more heterogeneous ($\beta \leq 3.5$), then regularization improves the performance of the algorithms; (2) if $\beta < 3$, then RSC and t-RSC outperform RSC_wp and SCP, verifying that the normalization step helps when the degrees are highly heterogeneous; and, finally, (3) uniformly across the setting of $\beta$, it is easier to cluster nodes with high leverage scores.

**Experiment 2.** This experiment compares SC, RSC, RSC_wp, t-RSC and SCP under the SBM with no degree heterogeneity. Each simulation has $K = 3$ blocks and $N = 1500$ nodes. As in the previous experiment, SNR is set to three. In this experiment, the average degree has three different settings: $10, 21, 30$. For each setting, the results are averaged over 50 samples of the network.

The right panel of Figure 2 shows the misclustering rate of SC and RSC for the three different values of the average degree. SCP, RSC_wp, t-RSC perform similarly to RSC, demonstrating that under the standard SBM (i.e. without degree heterogeneity) all spectral clustering methods perform comparably. The one exception is that under the sparsest model, SC is less stable than the other methods.

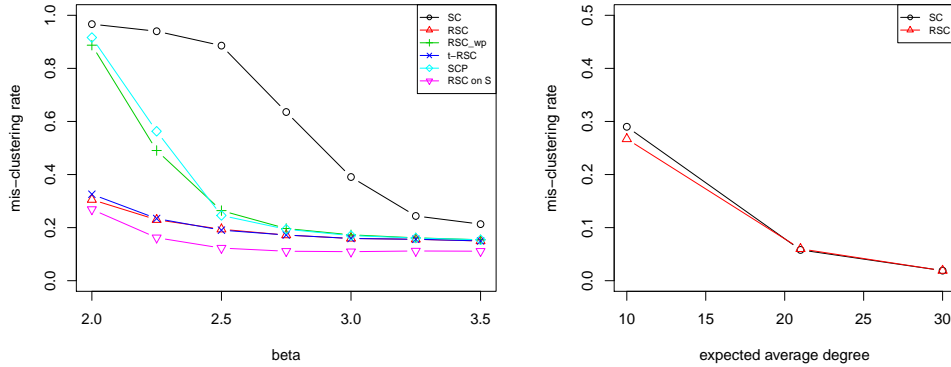

Figure 2: Left Panel: Comparison of Performance for SC, RSC, RSC_wp, t-RSC, SCP and (RSC on $S$) under different degree heterogeneity. Smaller $\beta$ corresponds to greater degree heterogeneity. Right Panel: Comparison of Performance for SC and RSC under SBM with different sparsity.

**Analysis of Blog Network.** This empirical network is comprised of political blogs during the 2004 US presidential election (Adamic and Glance [17]). Each blog has a known label as liberal or conservative. As in Karrer and Newman [10], we symmetrize the network and consider only the largest connected component of 1222 nodes. The average degree of the network is roughly 15. We apply RSC to the data set with $\tau$ ranging from 0 to 30. In the case where $\tau = 0$, it is standard Spectral Clustering. SC assigns 1144 out of 1222 nodes to the same block, failing to detect the ideological partition. RSC detects the partition, and its performance is insensitive to the $\tau$. With $\tau \in [1, 30]$, RSC misclusters $(80 \pm 2)$ nodes out of 1222.

If RSC is applied to the 90% of nodes with the largest leverage scores (i.e. excluding the nodes with the smallest leverage scores), then the misclustering rate among these high leverage nodes is $44/1100$, which is almost 50% lower. This illustrates how the leverage score corresponding to a node can gauge the strength of the clustering evidence for that node relative to the other nodes.

We tried to compare these results to the regularized algorithm in [1]. However, because there are several very small degree nodes in this data, the values computed in step 4 of the algorithm in [1] sometimes take negative values. Then, step 5 (b) cannot be performed.

# 6 Discussion

In this paper, we give theoretical, simulation, and empirical results that demonstrate how a simple adjustment to the standard spectral clustering algorithm can give dramatically better results for networks with heterogeneous degrees. Our theoretical results add to the current results by studying the regularization step in a more canonical version of the spectral clustering algorithm. Moreover, our main results require no assumptions on the minimum node degree. This is crucial because it allows us to study situations where several nodes have small leverage scores; in these situations, regularization is most beneficial. Finally, our results demonstrate that choosing a tuning parameter close to the average degree provides a balance between several competing objectives.

**Acknowledgements**

Thanks to Sara Fernandes-Taylor for helpful comments. Research of TQ is supported by NSF Grant DMS-0906818 and NIH Grant EY09946. Research of KR is supported by grants from WARF and NSF grant DMS-1309998.

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
