[Supplementary Material · supplementary material.pdf]

# 1 Supplementary materials for RSC

**Proof of Lemma 3.2:**

*Proof.* Recall that $\mathscr{D}_{ii} = \theta_i [D_B]_{z_i}$ and $[\Theta_\tau]_{ii} = \theta_i \frac{\mathscr{D}_{ii}}{\mathscr{D}_{ii}+\tau}$. The $ij$'th element of $\mathscr{L}_\tau$:

$$[\mathscr{L}_\tau]_{ij} = \frac{\mathscr{A}_{ij}}{\sqrt{(\mathscr{D}_{ii}+\tau)(\mathscr{D}_{jj}+\tau)}} = \frac{\theta_i\theta_j B_{z_iz_j}}{\sqrt{\mathscr{D}_{ii}\mathscr{D}_{jj}}} \sqrt{\frac{\mathscr{D}_{ii}}{\mathscr{D}_{ii}+\tau}\frac{\mathscr{D}_{jj}}{\mathscr{D}_{jj}+\tau}} = \frac{B_{z_iz_j}}{\sqrt{[D_B]_{z_i}[D_B]_{z_j}}} * \sqrt{[\Theta_\tau]_{ii}[\Theta_\tau]_{jj}}.$$

Hence,

$$\mathscr{L}_\tau = \Theta_\tau^{\frac{1}{2}} Z B_L Z^T \Theta_\tau^{\frac{1}{2}}.$$

$\square$

**Proof of Lemma 3.3:**

*Proof.* Let $C = (Z^T\Theta_\tau Z)^{1/2} B_L (Z^T\Theta_\tau Z)^{1/2}$. If $\theta_i > 0, i = 1, ..., N$, then $C \succ 0$ since $B \succ 0$ by assumption. Let $\lambda_1 \geq ... \geq \lambda_K > 0$ be the eigenvalues of $C$. Let $\Lambda \in \mathcal{R}^{K \times K}$ be a diagonal matrix with its $ss$'th element to be $\lambda_s$. Let $U \in \mathcal{R}^{K \times K}$ be an orthogonal matrix where its $s$'th column is the eigenvector of $C$ corresponding $\lambda_s, s = 1, ..., K$. By eigen-decomposition, we have $C = U\Lambda U^T$. Define $\mathscr{X}_\tau = \Theta_\tau^{\frac{1}{2}} Z (Z^T\Theta_\tau Z)^{-1/2} U$, then

$$\mathscr{X}_\tau^T \mathscr{X}_\tau = U^T (Z^T\Theta_\tau Z)^{-1/2}(Z^T\Theta_\tau Z)(Z^T\Theta_\tau Z)^{-1/2}U = U^T U = I.$$

On the other hand,

$$\mathscr{X}_\tau \Lambda \mathscr{X}_\tau^T = \Theta_\tau^{\frac{1}{2}} Z(Z^T\Theta_\tau Z)^{-1/2} C (Z^T\Theta_\tau Z)^{-1/2} Z^T \Theta_\tau^{\frac{1}{2}} = \Theta_\tau^{\frac{1}{2}} Z B_L Z^T \Theta_\tau^{\frac{1}{2}} = \mathscr{L}_\tau.$$

Hence, $\lambda_s, s = 1, ..., K$ are $\mathscr{L}_\tau$'s positive eigenvalues and $\mathscr{X}_\tau$ contains $\mathscr{L}_\tau$'s eigenvectors corresponding to its nonzero eigenvalues. For part 2, notice that $||\mathscr{X}_\tau^i||_2 = (\frac{[\Theta_\tau]_{ii}}{[Z^T\Theta_\tau Z]_{z_iz_i}})^{1/2}$, then

$$[\mathscr{X}_\tau^*]^i = \frac{\mathscr{X}_\tau^i}{||\mathscr{X}_\tau^i||_2} = \frac{(\frac{[\Theta_\tau]_{ii}}{[Z^T\Theta_\tau Z]_{z_iz_i}})^{1/2} Z_i U}{||\mathscr{X}_\tau^i||_2} = Z_i U.$$

Therefore, $\mathscr{X}_\tau^* = ZU$. $\square$

**Proof of Theorem 4.1:**

*Proof.* We extend the proof of Theorem 2 in Chung and Radcliffe [1] to the case of regularized graph laplacian. Let $H = \mathscr{D}_\tau^{-1/2} A \mathscr{D}_\tau^{-1/2}$. Then $||L_\tau - \mathscr{L}_\tau|| \leq ||H - \mathscr{L}_\tau|| + ||L_\tau - H||$. We bound the two terms separately.

For the first term, we apply the concentration inequality for matrix:

**Lemma 1.1.** *Let $X_1, X_2, ..., X_m$ be independent random $N \times N$ Hermitian matrices. Moreover, assunme that $||X_i - \mathbb{E}(X_i)|| \leq M$ for all $i$, and put $v^2 = ||\sum var(X_i)||$. Let $X = \sum X_i$. Then for any $a > 0$,*

$$pr(||X - \mathbb{E}(X)|| \geq a) \leq 2N \exp\left(-\frac{a^2}{2v^2 + 2Ma/3}\right).$$

Notice that $||H - \mathscr{L}_\tau|| = \mathscr{D}_\tau^{-1/2}(A - \mathscr{A})\mathscr{D}_\tau^{-1/2}$. Let $E^{ij} \in \mathcal{R}^{N \times N}$ be the matrix with 1 in the $ij$ and $ji$'th positions and 0 everywhere else. Let

$$X_{ij} = \mathscr{D}_\tau^{-1/2}((A_{ij} - p_{ij})E^{ij})\mathscr{D}_\tau^{-1/2}$$
$$= \frac{A_{ij} - p_{ij}}{\sqrt{(\mathscr{D}_{ii}+\tau)(\mathscr{D}_{jj}+\tau)}} E^{ij}.$$

$H - \mathscr{L}_\tau = \sum X_{ij}$. Then we can apply the matrix concentration theorem on $\{X_{ij}\}$. By similar argument as in [1], we have

$$||X_{ij}|| \leq [(\mathscr{D}_{ii} + \tau)(\mathscr{D}_{jj} + \tau)]^{-1/2} \leq \frac{1}{\delta + \tau}, \qquad v^2 = ||\sum E(X_{ij}^2)|| \leq \frac{1}{\delta + \tau}.$$

Take $a = \sqrt{\frac{3\ln(4N/\epsilon)}{\delta + \tau}}$. By assumption $\delta + \tau > 3\ln N + 3\ln(4/\epsilon)$, it implies $a < 1$. Applying Lemma 1.1, we have

$$pr(||H - \mathscr{L}_\tau|| \geq a) \leq 2N\exp\left(-\frac{\frac{3\ln(4N/\epsilon)}{\delta+\tau}}{2/(\delta+\tau) + 2a/[3(\delta+\tau)]}\right)$$

$$\leq 2N\exp(-\frac{3\ln(4N/\epsilon)}{3})$$

$$\leq \epsilon/2.$$

For the second term, first we apply the two sided concentration inequality for each $i$, (see for example Chung and Lu [2, chap. 2])

$$pr(|D_{ii} - \mathscr{D}_{ii}| \geq \lambda) \leq \exp\{-\frac{\lambda^2}{2\mathscr{D}_{ii}}\} + \exp\{-\frac{\lambda^2}{2\mathscr{D}_{ii} + \frac{2}{3}\lambda}\}$$

Let $\lambda = a(\mathscr{D}_{ii} + \tau)$, where $a$ is the same as in the first part.

$$pr(|D_{ii} - \mathscr{D}_{ii}| \geq a(\mathscr{D}_{ii} + \tau)) \leq \exp\{-\frac{a^2(\mathscr{D}_{ii} + \tau)^2}{2\mathscr{D}_{ii}}\} + \exp\{-\frac{a^2(\mathscr{D}_{ii} + \tau)^2}{2\mathscr{D}_{ii} + \frac{2}{3}a(\mathscr{D}_{ii} + \tau)}\}$$

$$\leq 2\exp\{-\frac{a^2(\mathscr{D}_{ii} + \tau)^2}{(2 + \frac{2}{3}a)(\mathscr{D}_{ii} + \tau)}\}$$

$$\leq 2\exp\{-\frac{a^2(\mathscr{D}_{ii} + \tau)}{3}\}$$

$$\leq 2\exp\{-\ln(4N/\epsilon)\frac{(\mathscr{D}_{ii} + \tau)}{\delta + \tau}\}$$

$$\leq 2\exp\{-\ln(4N/\epsilon)\}$$

$$\leq \epsilon/2N.$$

$$||\mathscr{D}_\tau^{-1/2}D_\tau^{1/2} - I|| = max_i|\sqrt{\frac{D_{ii} + \tau}{\mathscr{D}_{ii} + \tau}} - 1| \leq max_i|\frac{D_{ii} + \tau}{\mathscr{D}_{ii} + \tau} - 1|.$$

$$pr(||\mathscr{D}_\tau^{-1/2}D_\tau^{1/2} - I|| \geq a) \leq pr(max_i|\frac{D_{ii} + \tau}{\mathscr{D}_{ii} + \tau} - 1| \geq a)$$

$$\leq pr(\cup_i\{|(D_{ii} + \tau) - (\mathscr{D}_{ii} + \tau)| \geq b(\mathscr{D}_{ii} + \tau)\})$$

$$\leq \epsilon/2.$$

Note that $||L_\tau|| \leq 1$, therefore, with probability at least $1 - \epsilon/2$, we have

$$||L_\tau - H|| = ||D_\tau^{-1/2}AD_\tau^{-1/2} - \mathscr{D}_\tau^{-1/2}A\mathscr{D}_\tau^{-1/2}||$$

$$= ||L_\tau - \mathscr{D}_\tau^{-1/2}D_\tau^{1/2}L_\tau D_\tau^{1/2}\mathscr{D}_\tau^{-1/2}||$$

$$= ||(I - \mathscr{D}_\tau^{-1/2}D_\tau^{1/2})L_\tau D_\tau^{1/2}\mathscr{D}_\tau^{-1/2} + L_\tau(I - D_\tau^{1/2}\mathscr{D}_\tau^{-1/2})||$$

$$\leq ||\mathscr{D}_\tau^{-1/2}D_\tau^{1/2} - I||||\mathscr{D}_\tau^{-1/2}D_\tau^{1/2}|| + ||\mathscr{D}_\tau^{-1/2}D_\tau^{1/2} - I||$$

$$\leq a^2 + 2a.$$

Combining the two part, we have that with probability at least $1 - \epsilon$,

$$||L_\tau - \mathscr{L}_\tau|| \leq a^2 + 3a \leq 4a,$$

where $a = \sqrt{\frac{3\ln(4N/\epsilon)}{\delta + \tau}}$. $\qquad\qquad\qquad\qquad\qquad\qquad\qquad\qquad\qquad\qquad\square$

**Proof of Theorem 4.2:**

*Proof.* First we apply a lemma from McSherry [3]:

**Lemma 1.2.** *For any matrix $A$, let $P_A$ denotes the projection onto the span of $A$'s first $K$ left sigular vectors. Then $P_A A$ is the optimal rank $K$ approximation to $A$ in the following sense. For any rank $K$ matrix $X$, $||A - P_A A|| \leq ||L - X||$. Further, for any rank $K$ matrix $B$,*

$$||P_A A - B||_F^2 \leq 8K||A - B||^2. \tag{1}$$

Let $W \in \mathcal{R}^{K \times K}$ be a diagonal matrix that contains the K largest eigenvalues of $L_\tau$, $w_1 \geq w_2 \geq ... \geq w_K$. Let $\Lambda \in R^{K \times K}$ be the diagonal matrix that contains all positive eigenvalues of $\mathscr{L}_\tau$. Take $A = L_\tau$ and $B = \mathscr{L}_\tau$ in Lemma 1.2. then $P_{L_\tau} L_\tau = X_\tau W X_\tau^T$ and the previous inequality can be rewritten as

$$||P_{L_\tau} L_\tau - \mathscr{L}_\tau||_F^2 = ||X_\tau W X_\tau^T - \mathscr{X}_\tau \Lambda \mathscr{X}_\tau^T||_F^2 \leq 8K||L_\tau - \mathscr{L}_\tau||^2.$$

Then we apply a modified version of the Davis-Kahan theorem (Rohe et al. [4]) to $\mathscr{L}_\tau$.

**Proposition 1.3.** *Let $S \subset \mathcal{R}$ be an interval. Denote $\mathscr{X}_\tau$ as an orthonormal matrix whose column space is equal to the eigenspace of $\mathscr{L}_\tau$ corresponding to the eigenvalues in $\lambda_S(\mathscr{L}_\tau)$ (more formally, the column space of $\mathscr{X}_\tau$ is the image of the spectral projection of $\mathscr{L}_\tau$ induced by $\lambda_S(\mathscr{L}_\tau)$). Denote by $X_\tau$ the analogous quantity for $P_{L_\tau} L_\tau$. Define the distance between $S$ and the spectrum of $\mathscr{L}_\tau$ outside of $S$ as*

$$\Delta = \min\{|\lambda - s|; \lambda \text{ eigenvalue of } \mathscr{L}_\tau, \ \lambda \notin S, \ s \in S\}.$$

*if $\mathscr{X}_\tau$ and $X_\tau$ are of the same dimension, then there is an orthogonal matrix $\mathscr{O}$, that depends on $\mathscr{X}_\tau$ and $X_\tau$, such that*

$$||X_\tau - \mathscr{X}_\tau \mathscr{O}||_F^2 \leq \frac{2||P_{L_\tau} L_\tau - \mathscr{L}_\tau||_F^2}{\Delta^2}.$$

Take $S = (\lambda_K/2, 2)$, then $\Delta = \lambda_K/2$. By assumption (a) $\sqrt{\frac{K \ln(4N/\epsilon)}{\delta + \tau}} \leq \frac{1}{8\sqrt{3}} \lambda_K$, we have that when N is sufficiently large, with probability at least $1 - \epsilon$,

$$|\lambda_K - w_K| \leq ||L_\tau - \mathscr{L}_\tau|| \leq 4\sqrt{\frac{3 \ln(4N/\epsilon)}{\delta + \tau}} \leq \lambda_K/2.$$

Hence $w_K \in S$. $X$ and $\mathscr{X}$ are of the same dimension.

$$||X_\tau - \mathscr{X}_\tau \mathscr{O}||_F \leq \frac{\sqrt{2}||P_{L_\tau} L_\tau - \mathscr{L}_\tau||_F}{\Delta} \leq \frac{2\sqrt{2}||P_{L_\tau} L_\tau - \mathscr{L}_\tau||_F}{\lambda_K}$$

$$\leq \frac{8\sqrt{K}||L_\tau - \mathscr{L}_\tau||}{\lambda_K}$$

$$\leq \frac{C}{\lambda_K} \sqrt{\frac{K \ln(4N/\epsilon)}{\delta + \tau}}.$$

holds for $C = 32\sqrt{3}$ with probability at least $1 - \epsilon$.

For part 2, note that for any $i$,

$$||[X_\tau^*]^i - [\mathscr{X}_\tau^*]^i \mathscr{O}||_2 \leq \frac{||X_\tau^i - \mathscr{X}_\tau^i \mathscr{O}||_2}{min\{||X_\tau^i||_2, ||\mathscr{X}_\tau^i||_2\}},$$

We have that

$$||X_\tau^* - \mathscr{X}_\tau^* \mathscr{O}||_F \leq \frac{||X_\tau - \mathscr{X}_\tau \mathscr{O}||_F}{m},$$

where $m = min_i\{min\{||X_\tau^i||_2, ||\mathscr{X}_\tau^i||_2\}\}$.     □

**Proof of Main Theorem**

*Proof.* Recall that the set of misclustered nodes is defined as:

$$\mathcal{M} = \{i : \exists j \neq i, s.t. ||C_i \mathscr{O}^T - \mathcal{C}_i||_2 > ||C_i \mathscr{O}^T - \mathcal{C}_j||_2\}.$$

Note that Lemma 3.3 implies that the population centroid corresponding to $i$'th row of $\mathscr{X}_\tau^*$

$$\mathcal{C}_i = Z_i U.$$

Since all population centroids are of unit length and are orthogonal to each other, a simple calculation gives a sufficient condition for one observed centroid to be closest to the population centroid:

$$||C_i \mathscr{O}^T - \mathcal{C}_i||_2 < 1/\sqrt{2} \Rightarrow ||C_i \mathscr{O}^T - \mathcal{C}_i||_2 < ||C_i \mathscr{O}^T - \mathcal{C}_j||_2 \quad \forall Z_j \neq Z_i.$$

Define the following set of nodes that do not satisfy the sufficient condition,

$$\mathcal{U} = \{i : ||C_i \mathscr{O}^T - \mathcal{C}_i||_2 \geq 1/\sqrt{2}\}.$$

The mis-clustered nodes $\mathcal{M} \in \mathcal{U}$.

Define $Q \in \mathcal{R}^{N \times K}$, where the $i$'th row of $\mathbf{Q}$ is $C_i$, the observed centroid of node $i$ from k-means. By definition of k-means, we have

$$||X_\tau^* - Q||_2 \leq ||X_\tau^* - \mathscr{X}_\tau^* \mathscr{O}||_2.$$

By triangle inequality,

$$||Q - ZU\mathscr{O}||_2 = ||Q - \mathscr{X}_\tau^* \mathscr{O}||_2 \leq ||X_\tau^* - Q||_2 + ||X_\tau^* - \mathscr{X}_\tau^* \mathscr{O}||_2 \leq 2||X_\tau^* - \mathscr{X}_\tau^* \mathscr{O}||_2.$$

We have with probability at least $1 - \epsilon$,

$$\frac{|\mathcal{M}|}{N} \leq \frac{|\mathcal{U}|}{N} = \frac{1}{N} \sum_{i \in \mathcal{U}} 1$$

$$\leq \frac{2}{N} \sum_{i \in \mathcal{U}} ||C_i \mathscr{O}^T - \mathcal{C}_i||_2^2$$

$$= \frac{2}{N} \sum_{i \in \mathcal{U}} ||C_i - Z_i U \mathscr{O}||_2^2$$

$$\leq \frac{2}{N} ||Q - ZU\mathscr{O}||_F^2$$

$$\leq \frac{8}{N} ||X_\tau^* - \mathscr{X}_\tau^* \mathscr{O}||_F^2$$

$$\leq c_1 \frac{K \ln(N/\epsilon)}{Nm^2(\delta + \tau)\lambda_K^2}.$$

$\square$