[Reviews · NeurIPS 2013]

Submitted by Assigned_Reviewer_4

The authors present a theoretical analysis of spectral clustering in the degree corrected stochastic block model. Following earlier work of Chaudhuri et. al. they analyze spectral clustering with the regularized normalized Laplacian, first showing that the population Laplacian reflects the correct block structure, and then showing that the "noisy" graph Laplacian has a similar spectrum. Finally, they bound the number of mis-clustered nodes. They also consider an additional step of normalizing rows of the spectral embedding and show that the resulting bounds are related to the statistical leverage scores.

Overall the paper is well written (a few typos are pointed out at the end), addresses an interesting problem and provides a much simpler (albeit weaker) analysis as compared to the earlier work of Chaudhuri et. al.

Here are some issues that the authors should clarify:

1. I don't quite understand why the passage before equation (6) is an appropriate definition of mis-clustered nodes. There is already a natural definition from the ground truth clustering. It would help if the authors clarified the relation between the k-means 0/1 error and the set \mathcal{U}. Why are the nodes in \mathcal{U} the only ones on which k-means (which does not have access to population centroids, say with random initialization) makes errors? If not, it would help avoid confusion to not use the term "mis-clustering rate".

2. There should be a more detailed comparison to the earlier work of Chaudhuri et. al. In particular, they provide results on the parameters of the DC-SBM for which there are no errors made by their clustering algorithm and complement this with lower bounds. At least for the case of the SBM, rather than assuming (p-r) is a constant, a proper comparison to known results from McSherry/Chaudhuri et. al. should be made.

3. The role of normalization (of the embedding) is still not clear to me. While the figure shows a case where it seems to help in practice, it would help to clarify its role in the theory. Are the rates worse? If possible, the authors should compare the rates with and without the normalization, and if not they should clarify the reason for normalization.

4. Finally, in the experiments section - you report 50% improvement after dropping 10% of the nodes. However, although the algorithm you propose clusters *all* nodes you only report accuracy on the retained nodes. Does the accuracy on the full set improve at all?

Minor typos:

1. Two statistical model --> models
2. Line 59 -- we study **an**
3. 102 -- emphasis --> emphasize
4. 243 and 244 -- centroid --> centroids
Summary: The paper provides an interesting theoretical analysis for a (modification of a) widely used clustering algorithm. I have a few issues with the theoretical results presented but I recommend acceptance.

Submitted by Assigned_Reviewer_5

NA
Summary: NA

Submitted by Assigned_Reviewer_6

Summary:
This paper analyses spectral clustering schemes where a small positive constant (\tau) is added to the node degrees as a form of regularization. The main contributions are the removal of a minimum degree assumption common in previous studies and some insights on selecting a range of good values for \tau.

Quality:
+ The theoretical part of the paper seems OK and convincing in most parts.
+ The few experiments performed are evaluated in terms of appropriate unsupervised measures such as the misclustering rate.
- The experimental section is rather weak with only one real-life dataset.
- The contributions does not seem very significant and it is unclear why someone would prefer the proposed scheme to similar ones.
- No out-of-sample extensions are provided.
- It is claimed that the paper provides insight on the "star-shaped" figure that appears in the empirical eigenvectors. However, these insights are not explicitly mentioned in the manuscript. The "star" shape is known to appear in ideal conditions for the top eigenvectors of D^{-1/2}AD^{-1/2} due to the normalization while the top eigenvectors of D^{1}A are known to appear localized (piecewise constant). What are the insights provided by the proposed scheme ?

Clarity:
+ On average, the paper is OK to read with the following exceptions:
- How are the \theta parameters computed ?
- Do the results hold for weighted graphs? Whether yes or no, it needs to be clearly explained in the manuscript.

Originality:
+ The theoretical analysis about selecting good values for \tau seems novel.
- The paper relies heavily on previous work and the contributions are very incremental and unlikely to be significant.

Significance:
- The main contributions are rather weak. It is not clear how the lack of the minimum degree assumption common in previous studies is significant here. The influence of the proposed value for \tau on the clustering results is not properly studied either.
Summary: OK paper showing theoretical analysis on the benefits of regularization in spectral clustering. The main contributions are on the weak side and unlikely to be significant.
Author Feedback

Author rebuttal: Thank you to the reviewers for their thoughtful comments.

We emphasize the following summary from Reviewer_5:
"The main contribution of the paper is to start to deal with a very awkward assumption that has appeared in related prior work and much of the related theoretical work on graphs with heavy-tailed degree distributions, namely that for analytical convenience that work assumes that the minimum degree in the graph is quite large. . . That assumption is strongly violated in many realistic graphs, rendering prior work less than applicable. When that assumption is removed, one gets localization on the eigenvectors."

This paper provides the first understanding of regularized eigenvectors for low degree nodes. Importantly, we found
(a) that the regularization is most helpful in regimes that the previous analysis failed to address (i.e. low degree nodes).
(b) the leverage scores provide a preliminary measure of statistical uncertainty. Previous estimation results have not provided insight at this level of granularity; instead, previous results have studied the global estimation performance.

Item specific responses are below.

Reviewer_4 --"need a more detailed comparison with the work of Chaudhuri et al."
Our results compare very favorably in simulation because their estimator is often not computable. In the theory, the results are mixed. We will rework the introduction to clarify the following points:
(Simulations) In step 4 of their algorithm, Chaudhuri et al. define a threshold \lambda_u for each node u. In simulation, these \lambda's are often less than 0, which makes step 5 not computable. To guarantee that \lambda_u >0, we must have degree(u) > 64 ln(6*n / \delta). Here delta is the probability threshold in their main theorem.

(Theory) Our theoretical results have the advantage of not assuming anything about the minimum degree. As Reviewer_5 observes, this is of fundamental importance. The disadvantage of our theoretical results is that we do not prove perfect clustering. This technical difficulty appears in all spectral clustering results that study a canonical version of the spectral clustering (eigendecomposition + k-means). Chaudhuri et al. circumvent this difficulty by using half the nodes to compute singular vectors and cluster the other half of the nodes.

Reviewer_4 --"why normalize the rows of eigenvectors?"
In a highly cited paper, Ng, Jordan and Weiss (NIPS 2002) proposed normalizing the rows of eigenvectors and their motivation comes from matrix perturbation theory. We provide the first statistical analysis of this step and in fact the analysis nicely pairs with the Degree Corrected Stochastic Blockmodel. Our revised simulation section (discussed below) demonstrates that normalizing aids estimation when the networks have degree heterogeneity.

Reviewer_4 --"Finally, in the experiments section - you report 50% improvement after dropping 10% of the nodes. However, although the algorithm you propose clusters *all* nodes you only report accuracy on the retained nodes. Does the accuracy on the full set improve at all?"
Compared with SC, it does. The data analysis of the blog network illustrates two points:
(1) RSC is significantly better than SC on the whole dataset.
(2) The performance of RSC is more reliable on nodes with high leverage scores, demonstrating how leverage scores provide a rough measure of statistical uncertainty at a node-level-resolution.

Reviewer_4 --"Is the definition of mis-clustered nodes appropriate?"
Because cluster labels are not identifiable, there is no canonical measure of "misclustered". We have adopted the definition from Rohe, Chatterjee, and Yu (AoS 2011) because it is suitable for k-means. We will include it in the revised supplementary material.

Reviewer_4 --"typos"
Thank you for finding and reporting the typos. We have corrected them in the revised paper.

Reviewer_5 --"move the remarks on lines … to more prominent position."
In the revised paper, we will reorganize the material to place greater emphasize on the following points and how they relate to each other:
a) Remove the assumption on lowest expected degree
b) Relation between leverage score and clustering performance
c) Graph sparsity and localization on eigenvectors

Reviewer_5 and 6 -- Empirical evaluation is limited.
We have revised experiment 2 to more directly investigate the role of the degree distribution. We simulate graphs with power law degree distributions with shape parameter ranging in {4, 3.5, 3, 2.5, 2} and fixing the expected average degree to be 8. These simulations demonstrate
(1) the instability of standard SC,
(2) how normalization helps when shape parameter is less than 3, and
(3) how uniformly across simulations it is easier to cluster the nodes with high leverage scores.
These findings are consistent with our theoretical results on introducing regularization term and normalizing.

Reviewer_6 --"What are the insights on the ''star-shaped" pattern?"
We give an model based explanation of the "star shaped" pattern. We show that under the DC-SBM, or the extended planted partition model, the population top eigenvectors shows exact "star shape" (see figure 1 left panel.). And our theoretical result shows that the empirical top eigenvectors are close to these population ones.
We will add detailed explanation in the DC-SBM section in the revised paper.

Reviewer_6 --"How are the \theta parameters computed?"
In simulation, we generate the \theta parameters from power-law distribution. We will clarify this in the paper.

Reviewer_6 --"Do the results hold for weighted graphs?"
This is an interesting and relevant question. We suspect that it does, but due to space constraints we cannot address the issue here.